# Structural and functional characterization of TraI from pKM101 reveals basis for DNA processing

Annika Breidenstein[1,2] , Josy ter Beek[1,2] , Ronnie P-A Berntsson[1,2]

Type 4 secretion systems are large and versatile protein machineries that facilitate the spread of antibiotic resistance and other virulence factors via horizontal gene transfer. Conjugative type 4 secretion systems depend on relaxases to process the DNA in preparation for transport. TraI from the well-studied conjugative plasmid pKM101 is one such relaxase. Here, we report the crystal structure of the trans-esterase domain of TraI in complex with its substrate *oriT* DNA, highlighting the conserved DNA-binding mechanism of conjugative relaxases. In addition, we present an apo structure of the trans-esterase domain of TraI that includes most of the flexible thumb region. This allows us for the first time to visualize the large conformational change of the thumb subdomain upon DNA binding. We also characterize the DNA binding, nicking, and religation activity of the trans-esterase domain, helicase domain, and full-length TraI. Unlike previous indications in the literature, our results reveal that the TraI trans-esterase domain from pKM101 behaves in a conserved manner with its homologs from the R388 and F plasmids.

## Introduction

Antibiotic resistance is one of the most pressing health challenges in today's world. One of the main drivers of this worldwide problem is the ability of pathogens to spread resistance genes using horizontal gene transfer. This is a process that is often facilitated by conjugative type 4 secretion systems (T4SSs) (Juhas, 2015; Gonzalez-Rivera et al, 2016). These are megadalton-sized, multicomponent systems that transport DNA and proteins from a bacterial donor cell into a recipient cell (Grohmann et al, 2018; Waksman, 2019). In Gram-negative bacteria, T4SS consist of a channel that is made up of an inner membrane complex that is connected with an outer-membrane core complex by a stalk (Low et al, 2014; Chandran Darbari & Waksman, 2015; Costa et al, 2021). Other important features are the pilus, which extends out from the cell and is involved in mediating attachment to recipient cells, and the type IV coupling protein, an ATPase that helps to recruit the single-stranded DNA substrate which is coupled to a relaxase

(Arutyunov & Frost, 2013; Grohmann et al, 2018). In recent years, researchers have generated structural and functional insight into these systems, especially from Gram-negative (G−) bacteria, among others the R388 plasmid and the pKM101 system (Khara et al, 2021; Macé et al, 2022). For a recent review of the structure and function of T4SSs, please see Costa et al (2021). The focus on this work is the T4SS of pKM101, which belongs to the class of minimal T4SS that consist of 12 proteins homologous to the paradigmatic VirB/VirD4 T4SS from *Agrobacterium tumefaciens* (Christie, 2004, 2016).

For all known T4SSs that transfer conjugative plasmids, the plasmid DNA must be processed before it can be transported. This is done via the relaxosome complex, which consists of a relaxase, accessory factor protein(s), and the DNA (Grohmann et al, 2018). To form this complex, one or several accessory proteins bind to the origin of transfer (*oriT*) on the DNA and locally melt the double-stranded DNA to promote relaxase binding to a defined sequence, often forming a hairpin, close to the nicking site (Zechner et al, 2017). The relaxase binds this single-stranded *oriT* DNA via its N-terminal trans-esterase domain (Garcillán-Barcia et al, 2009). This domain reacts with the DNA via a transesterification reaction at the specific *nic*-site, which generates the transfer intermediate consisting of the relaxase covalently bound to the 5' end of the cleaved transfer strand (T-strand) (Byrd & Matson, 1997). Many relaxases have a second functional domain in the more variable C-terminal part of the enzyme. This is often a helicase domain, which unwinds the DNA to allow for transport of the single-stranded transfer-strand DNA (Garcillán-Barcia et al, 2009). The relaxase–transfer-strand complex (T-complex) is recruited to the T4SS by the type IV coupling protein and is transported through the T4SS channel into the recipient cell (Alvarez-Martinez & Christie, 2009). Once present in the recipient cell, the trans-esterase domain religates the DNA to regenerate the circularized plasmid (Waksman, 2019).

Relaxases have been phylogenetically classified into eight MOB families: $MOB_F$, $MOB_H$, $MOB_Q$, $MOB_C$, $MOB_P$, $MOB_V$, $MOB_T$, and $MOB_B$. Of these, pKM101-encoded TraI ($TraI_{pKM101}$) belongs to the $MOB_{F11}$ subclade together with its closest relative TrwC from plasmid R388 ($TrwC_{R388}$) and the identical TraI from the sister plasmid pCU1 ($TraI_{pCU1}$) (Paterson et al, 1999; Garcillán-Barcia et al, 2009; Guglielmini et al, 2011). Although structural information is available

[1]Department of Medical Biochemistry and Biophysics, Umeå University, Umeå, Sweden   [2]Wallenberg Centre for Molecular Medicine, Umeå University, Umeå, Sweden

Correspondence: josy.beek@umu.se; ronnie.berntsson@umu.se

for the trans-esterase domains of several MOB_F relaxases (De La Cruz et al, 2010; Zechner et al, 2017), structural data for the substrate DNA-bound state is only available for TrwC_R388 and TraI from the F-plasmid (TraI_F), which belongs to subclade MOB_F12 (Larkin et al, 2005; Boer et al, 2006). Furthermore, there is only very limited data on full-length relaxases, again mostly from TraI_F. TraI_F consists of one trans-esterase domain, a vestigial helicase domain, and an active helicase domain (Fig 1A) and binds *oriT* DNA as a heterogenous dimer. One TraI_F monomer adapts an open conformation and binds *oriT* at a hairpin positioned 5′ of the *nic*-site with the trans-esterase domain, whereas a second TraI_F binds ssDNA with the helicase domains in a closed conformation. These two states are incompatible with each other in a single monomer (Ilangovan et al, 2017). A cryo-EM structure of the helicase-bound state is available and shows how the single-stranded DNA is almost entirely surrounded by the helicase domains. This structure is of full-length TraI_F and visualizes all of the protein except the very flexible C-terminal domain (Ilangovan et al, 2017). It is not clear to what extent this structure and mode of action apply to TraI_pKM101 as it is much shorter and completely lacks the vestigial helicase domain (Fig 1A).

In this study, we present the crystal structure of the trans-esterase domain of TraI_pKM101 bound to its substrate *oriT* DNA. This structure confirms the highly conserved DNA-binding mode of the trans-esterase domain observed in other MOB_F family relaxases. We also present the apo structure with the flexible thumb-subdomain defined, which shows a large conformational change between the apo and substrate-bound structures. TraI is further characterized via electrophoretic mobility shift assays (EMSAs) and nicking and religation assays.

## Results

### Purification of TraI and its functional domains

TraI and its functional domains as predicted by PsiPred (Buchan & Jones, 2019), the trans-esterase domain (TraI-TE) and the helicase domain (TraI-H), were expressed in *E. coli* with a cleavable His-tag and purified to homogeneity in a three-step process. TraI, TraI-TE, and TraI-H, all with their His-tags cleaved off, were shown to be monomeric in solution by size-exclusion chromatography coupled to multi-angle light scattering (SEC-MALS) (Fig S1). It is interesting to note that full-length TraI was initially purified without removing the His-tag (TraI_His) and was then shown to be in an oligomer–monomer equilibrium, with the molecular weight of the oligomer determined to be 485±10 kD, which corresponds well to the expected molecular weight of a tetramer (492 kD) (Fig S2). Because we only observed oligomerization of TraI_His, but not for TraI without the His-tag or TraI-TE and TraI-H, we conclude that TraI is a monomer and that the initially observed oligomerization of the full-length protein was an artifact induced by the His-tag.

### TraI is a functional relaxase capable of binding, nicking, and religating *oriT*-DNA

To biochemically characterize TraI and its functional domains, we performed several activity assays. To estimate the differences in binding affinity and specificity of the different TraI domains, we used EMSAs. We investigated DNA binding to a fluorescent 22-mer with the post-*nic* DNA sequence from the *oriT* of pKM101 (*oriT*22-F), a fluorescent 35-mer of the pre-*nic* DNA containing the hairpin structure (F-*oriT*35), and a fluorescent full-length 57-mer with both the pre- and post-*nic* sequence (F-*oriT*57), as well as randomly sequenced 57- and 35-mers (Fig 1B and Table S1).

We found that the TraI trans-esterase domain binds both F-*oriT*57 and F-*oriT*35 DNA with high affinity (at 200 nM over half of the DNA is bound, Figs 2A and S3A), whereas it only interacts with post-*nic* or random DNA at protein concentrations above 800 or 1,600 nM (Figs 2A and S3A). This indicates that the trans-esterase domain binds sequence-specific to the pre-*nic* region. In contrast, the TraI helicase domain shows only low affinity binding to both *oriT* and random DNA and seems to have a stronger interaction with the longer 57-mer than with the 35- and 22-mer DNA constructs (Figs 2B and S3B). This domain thus appears to have no sequence specificity and to prefer DNA fragments longer than 35 nucleotides.

As full-length TraI includes both the helicase and the trans-esterase domains, we expected to find a combination of their DNA-binding capabilities. TraI was indeed found to bind both *oriT* and random DNA. Interestingly, the random DNA shows a significant mobility shift with TraI protein concentrations of 200 nM and above (Fig 2C), whereas 400 nM is needed when the TraI helicase domain is used in isolation (Fig 2B). Both F-*oriT*57 and F-*oriT*35 show a complete mobility shift with TraI concentrations of 200 nM and

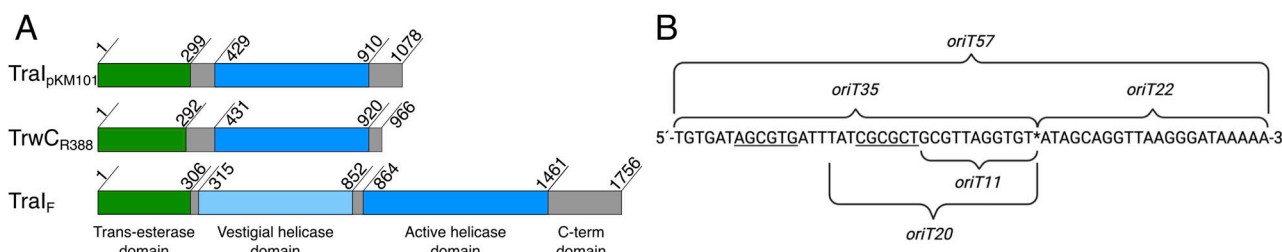

**Figure 1. Schematic overviews of domain organization of relaxases and the cognate *oriT* of pKM101.**
**(A)** Domain structure of three MOB_F family relaxases TraI_pKM101 (identical to TraI_pCU1), TrwC_R388, and TraI_F (Ilangovan et al, 2017; Zechner et al, 2017). Trans-esterase domains are shown in green, vestigial helicase domain in light blue, active helicase domains in dark blue, and the C-terminal domains and linker regions are shown in gray. **(B)** *oriT* sequence of pKM101 and an overview of the different *oriT* nucleotides used in this study. * indicates the *nic*-site, underlined regions of the sequence indicate the bases participating in hairpin formation.

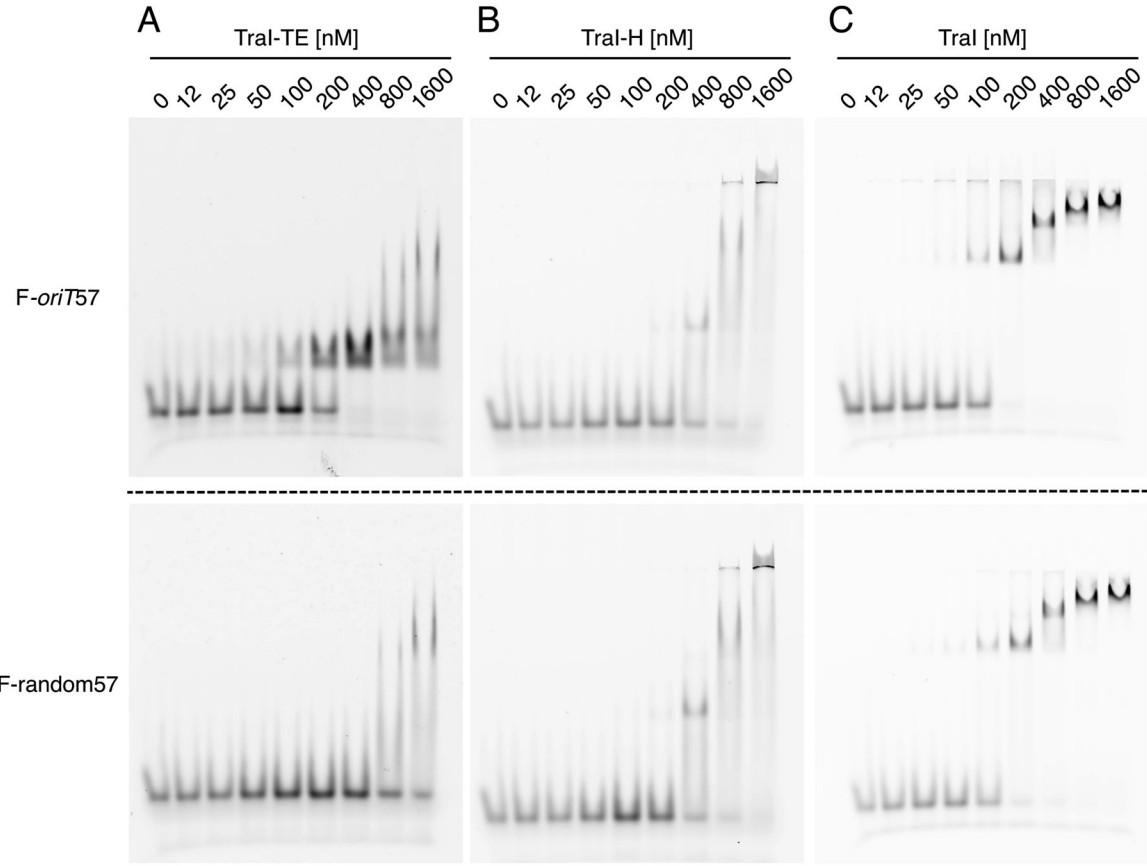

**Figure 2. TraI binding to *oriT* or random DNA.**
**(A, B, C)** Electromobility shift assay using 50 nM fluorescent 57-mer *oriT* (F-*oriT*57) (upper panel) or random DNA (lower panel) of the trans-esterase domain (TraI-TE) (A), helicase domain (TraI-H) (B), or full-length TraI (C).

above (Figs 2C and S3C), compared with 400 nM for the trans-esterase domain (Figs 2A and S3A). Both observations therefore indicate a significant increase in the apparent affinity of the full-length TraI as compared with its individual domains. At higher TraI concentrations, additional supershifts are observed (Fig 2C). An additional observation was an initial very high shift that was observed for F-oriT57 and F-oriT35 already with 50 nM TraI (Figs 2C and S3C) (similar to the high shift seen at the highest concentration of TraI-H [Figs 2B and S3B]). However, this high–molecular-weight band was only observed up to 400 nM TraI and disappears as the protein concentration is increased further. This result has been consistently observed in mobility shift assays with *oriT* DNA (both *oriT*35 and *oriT*57) but not with random DNA (Figs 2C and S3C).

We also examined the capacity of TraI to nick and religate its substrate *oriT* DNA. TraI was incubated with the fluorescent full-length F-*oriT*57, containing the recognition hairpin, the *nic*-site, and a post-*nic* stretch of 22 bases, or a random DNA sequence of the same length. DNA products were consequently separated on a denaturing polyacrylamide gel. Our results (Fig 3A and C) demonstrate that both TraI and TraI-TE can nick about 50% of the tested *oriT* DNA substrate, whereas neither nicked the random DNA control under the same conditions. We were unable to detect a higher degree of cleaved F-*oriT*57 DNA and predicted that this was because of the expected religation ability of TraI. To test this, we added a

shorter version of the fluorescent pre-*nic* DNA (F-*oriT*20) to the incubation mixture of TraI. In this experiment, F-*oriT*57 is expected to be (partly) cleaved by TraI into a fluorescent 35-mer and a non-fluorescent 22-mer that stays covalently bound to TraI. If religation occurs, the 22-mer could either be religated to the fluorescent 35-mer to re-form the original F-*oriT*57 or to the F-*oriT*20 to form a fluorescent 42-mer. This 42-mer was indeed seen on the denaturing gel (Fig 3B and D, left-side). This nicking and religation activity of TraI was further investigated by incubating the protein with a fluorescent F-*oriT*35 and a non-fluorescent *oriT*57. The appearance of F-*oriT*57 indicates that non-fluorescent *oriT*57 was nicked and that the resulting 22-mer was ligated to F-*oriT*35 (Fig 3B and D, right-side).

**Crystal structure of TraI trans-esterase domain in complex with *oriT* DNA**

To obtain a mechanistic understanding of the sequence-specific DNA-binding ability of TraI, purified TraI-TE was co-crystallized with single-stranded 11-mer *oriT* DNA (*oriT*11) (Fig 1B), and the obtained crystal structure was refined at a resolution of 2.1 Å. Virtually all residues were visible in the electron density, apart from five residues in flexible loop regions and at the C-terminus. We were also able to model 10 of the 11 bases into the electron density of the bound DNA (Fig 4).

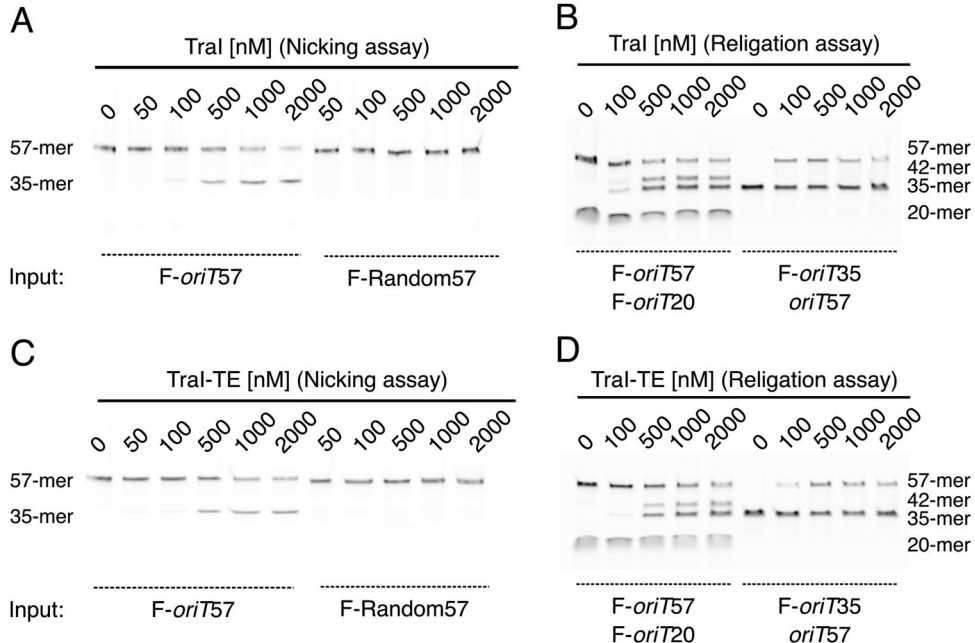

**Figure 3. DNA nicking and religation by TraI. (A, B, C, D)** Activity assays of TraI (A, B) and TraI-TE (C, D). **(A, C)** Nicking assays. F-*oriT*57 (left) and F-Random57 (right) were incubated with increasing protein concentrations, and the DNA was separated on a denaturing gel, showing the emergence of a 35-mer nicking product for *oriT* DNA only. **(B, D)** Religation assays. F-*oriT*57 and F-*oriT*20 were incubated with increasing protein concentrations (left-side). The appearance of a 35-mer product indicates nicking of F-*oriT*57, and the appearance of a 42-mer product indicates religation of the covalently bound post-nick DNA with the shorter F-*oriT*20 pre-nick DNA. F-*oriT*35 and *oriT*57 were incubated with increasing protein concentrations (right-side). The appearance of a 57-mer product shows nicking of the non-fluorescent *oriT*57 and religation with the fluorescent F-*oriT*35.

The overall architecture of TraI-TE can be described as a hand, with the bound DNA located between the palm and the C-terminal thumb subdomains (Figs 4A and S4A). The determined structure is similar to the previously described homologs TrwC$_{R388}$ and TraI$_F$ (Fig 4B) (Guasch et al, 2003; Larkin et al, 2005; Boer et al, 2006). A conserved histidine triad coordinates a divalent metal that is required for the trans-esterase reaction in the homologs (Byrd & Matson, 1997; Datta et al, 2003; Boer et al, 2006). In TraI, this triad is formed by H149, H160, and H162 (Fig 4C). Electron density was observed in the middle of this triad, and refinement against the possible metal ions yielded the best fit for manganese (Fig S4B and C), albeit this does not rule out that other divalent cations are bound. The conserved aspartate, D84 in TraI$_{pKM101}$, is situated within hydrogen-bonding distance of both H162 and Y18 (2.8 and 2.7 Å, respectively, Fig 4C) as earlier described for TrwC$_{R388}$ and TraI$_F$ (Larkin et al, 2005; Boer et al, 2006). It is therefore in position to activate the catalytic tyrosine Y18, the first tyrosine of the conserved YY-X$_{(5–6)}$–YY motif (Garcillán-Barcia et al, 2009; Zechner et al, 2017), which is predicted to occur during the trans-esterification reaction with the scissile phosphate at the *oriT* nic-site.

The DNA substrate used for co-crystallization was *oriT*11, representing the 11 bases directly 5′ of the nic-site. Of these, the first base at the 5′ end T1 did not yield any defined electron density, likely because it was not properly stabilized (Fig 4D). Although the first visible base C2 does not show any specific contacts with TraI-TE, the following guanine (G3) forms four hydrogen bonds with R80 and D182 using both the Hoogsteen and Watson–Crick edge, and the following thymine (T4) forms two hydrogen bonds with N181. The following two bases, T5 and A6, are stacked on top of each other perpendicular to the orientation of the previous three bases and form hydrogen bonds with K188 (T5) and Q256 (A6). These initial bases are oriented in a position that is pointing away from the active site. The following bases place the DNA in a conformation

resembling a U-turn that locates the scissile phosphate of T11 in close proximity to the catalytic tyrosine. This bent DNA conformation is stabilized with multiple hydrogen bonds, combined with hydrophobic and pi–pi interactions between the stacked bases G8 and G10. G7 forms two hydrogen bonds with L2 at its Watson–Crick edge and an additional one with K196 from its Hoogsteen edge. G8 forms two Watson–Crick hydrogen bonds with D3, and G10 forms two Hoogsten hydrogen bonds with R250. The phosphates of the three most 3′ bases are also stabilized with several hydrogen bonds (S236 for T9, S236 and K91 for G10, and R235 and R153 for T11) (Fig 4D). Taken together, these DNA–DNA and DNA–protein interactions illustrate a binding mechanism that relies both on the secondary structure and the specific sequence of the DNA.

**Apo structure reveals rearrangement of the thumb subdomain upon DNA binding**

An important feature of the DNA binding is the TraI-TE C-terminal thumb subdomain, consisting of two α-helices connected by a loop region (Figs 4A and S4A). This subdomain lies on top of the 3′ part of the DNA and forms hydrogen bonds with the *oriT* DNA via residues R235, R250, and Q256 (Fig 4D). The thumb subdomain is highly flexible and has previously only been observed in relaxase structures with DNA bound. Unexpectedly, our co-crystallization experiment resulted in an additional crystal with the space group $P2_12_12_1$ that resolved to 1.7 Å. This structure had a different unit cell, which had the fortuitous outcome of supporting an apo state of TraI-TE, in which we were able to build most of the thumb subdomain in an open conformation (Fig 5A). The electron density corresponding to residues 261–271 was of insufficient quality to build these residues in the model, but inspection of the density suggests that the α-helix continues toward the base of the thumb. Between the apo- and the DNA-bound state, the thumb moves up to

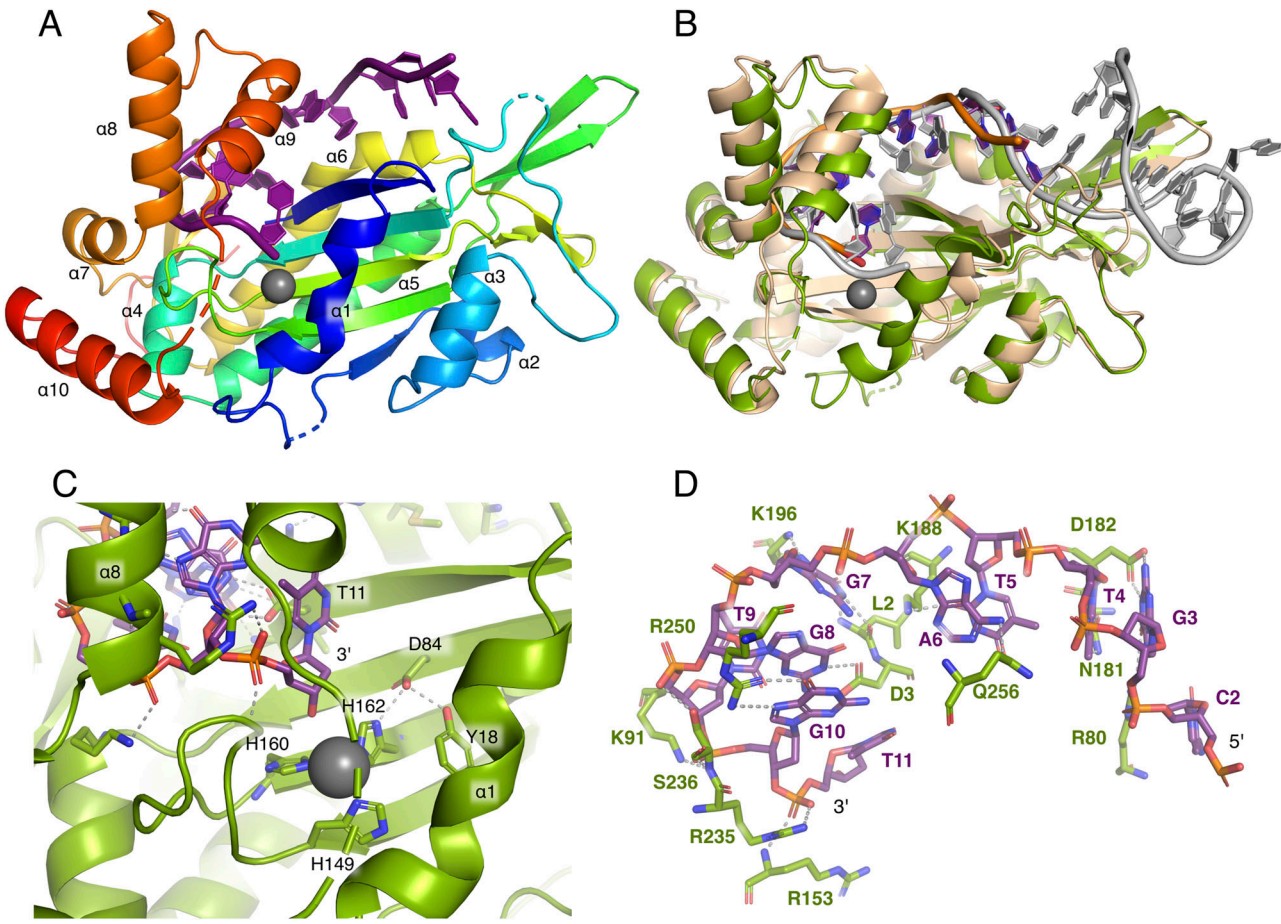

**Figure 4. Crystal structure of DNA-bound trans-esterase domain of TraI$_{pKM101}$.**
In all panels, TraI is colored green, the DNA bound by TraI$_{pKM101}$ is shown in purple and orange, and the bound divalent cation (here modeled as Mn$^{2+}$) in gray, unless otherwise indicated. **(A)** Cartoon representation of the trans-esterase domains of TraI$_{pKM101}$ colored from blue at the N-terminal to red at the C-terminal, with α-helices labeled from 1 to 10. **(B)** Superimposition of the trans-esterase domains of TraI$_{pKM101}$ and TrwC$_{R388}$ (protein in beige, DNA in gray, PDB: 2CDM). **(C)** Active site of TraI-TE$_{pKM101}$ consisting of the histidine triad coordinating a Mn$^{2+}$ (H149, H160, and H162), conserved aspartate (D84), and catalytic tyrosine (Y18). **(D)** Bound *oriT* DNA and its interactions with TraI-TE. Residues that form hydrogen bonds with DNA and the DNA itself are represented as sticks.

40 Å and undergoes important structural changes. The observed conformational differences include a reorganization of α-helix 8 and the extension of α-helix 9 (numbered based on the DNA-bound structure as one helix disappears in the apo structure) to transition from the apo to the DNA-bound structure, as illustrated in Fig 5A and Video 1. These rearrangements reveal the structural changes that are necessary to first allow DNA to enter the binding site and to subsequently keep the DNA in place, highlighting the importance of the thumb subdomain in this process. The C-terminal helix at the base of the thumb subdomain (α-helix 10) interacts with the loop at the end of α-helix 1 via a hydrogen bond between R274 and D25 in both the DNA-bound and apo structures (Fig S5A). An additional interaction is made between H278 and Y26, the third tyrosine of the YY-X$_{(5-6)}$–YY motif, but only in the DNA-bound structure.

Both the apo and the DNA-bound structure of TraI-TE presented here show differences when compared to the available structure from TraI$_{pCU1}$, even though the domains have an identical primary sequence (Fig 5B, PDB: 3L6T) (Nash et al, 2010). In TraI$_{pCU1}$, the C-terminal part of the protein containing the thumb subdomain

and the base of the thumb are missing, and the loop at the end of α-helix 1 is pointing away from the active site, possibly because it is not stabilized by the missing α-helix 10 (Fig 5B).

## Discussion

In this study, we structurally and biochemically characterized the TraI relaxase from the MOB$_F$ family plasmid pKM101 to better understand the diversity of the relaxases in this protein family as they are of key importance to type IV secretion.

We performed mobility shift assays (Figs 2 and S3) to estimate the DNA-binding affinity of full-length TraI and to compare this to the affinity of the trans-esterase (TraI-TE) and helicase (TraI-H) domains individually. Both full-length TraI and the helicase domain bound DNA in a sequence-unspecific manner as both bound to a similar degree to *oriT* and random DNA (Figs 2 and S3). This is as expected because these helicases are predicted to unwind the

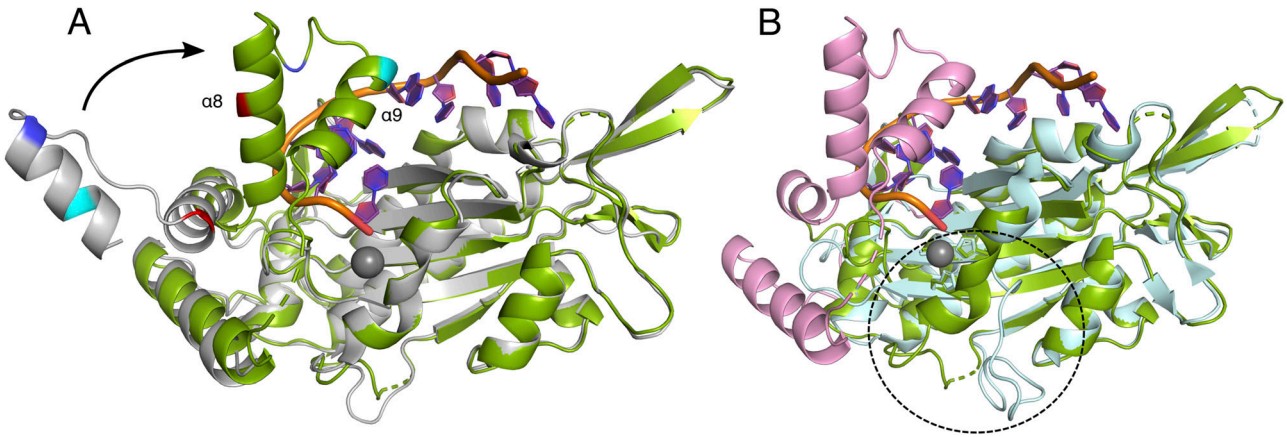

**Figure 5. Overview of the conformational changes between the apo and the DNA-bound structures and the differences with the previously published structure of the identical TraI-TE$_{pCU1}$.**
**(A)** DNA-bound structure of TraI$_{pKM101}$ (protein in green, DNA in purple and orange, Mn$^{2+}$ as a gray sphere), superimposed with the apo structure (light gray). To provide a visual aid in understanding the conformational changes that occur in the thumb subdomain upon DNA binding, residue K241 (red), R251 (blue), and Q256 (cyan) are highlighted in both structures, and α-helix 8 and 9 are indicated in the DNA-bound structure. **(A, B)** DNA-bound structure of TraI$_{pKM101}$ (colored as in panel (A)) with the thumb subdomain highlighted in pink, superimposed with the previously published structure of TraI-TE$_{pCU1}$ (cyan, PDB: 3L6T). TraI-TE$_{pCU1}$ lacks the thumb subdomain, and differences with TraI-TE$_{pKM101}$ are also observed close to the active site, indicated by the dashed circle.

entire conjugative plasmid and therefore should not have any sequence specificity. In contrast, the trans-esterase domain of TraI (TraI-TE) showed a higher affinity for *oriT* DNA as compared with random DNA (Figs 2 and S3). Although the mobility shift assays presented here are not suitable for precise $K_D$ determinations, the apparent $K_D$ of the trans-esterase domain of TraI$_{pKM101}$ for *oriT* DNA is <200 nM as more than half of the DNA was shifted upward at 200 nM (Fig 2A). This binding is sequence-specific as interactions with random DNA were only observed at protein concentrations above 800 nM (Fig 2A). Nicking and religation assays show that full-length TraI and trans-esterase domain (TraI-TE) are both equally capable of nicking and religating the *oriT* DNA but not random DNA. This shows that the trans-esterase domain is active on its own and confirms its *oriT* specificity (Fig 3).

The observed affinity and specificity of the trans-esterase domain for its cognate *oriT* DNA is a feature that has been described in various TraI$_{pKM101}$ homologs, such as TrwC$_{R388}$ and TraI$_F$ (Stern & Schildbach, 2001; Guasch et al, 2003; Carballeira et al, 2014). However, it is contradictory to a previous report on the trans-esterase domain of TraI$_{pCU1}$ (Nash et al, 2010). pCU1 is an antibiotic resistance plasmid that shares a common ancestor with pKM101 and R46 (Paterson et al, 1999). The pKM101 and pCU1 conjugative plasmids have an identical *oriT* region and encode identical relaxases: TraI$_{pKM101}$/TraI$_{pCU1}$. In spite of this, the trans-esterase domain of TraI$_{pCU1}$ was previously described to bind DNA in a weak and sequence-independent manner, with $K_D$ values for binding to *oriT* and non-*oriT* oligomers ranging from 0.7 to 1.2 μM (Nash et al, 2010). The DNA binding of TraI$_{pKM101}$ and the relaxase domain of TraI$_{pCU1}$ was measured using different methods and using slightly different protein constructs. The differences in the protein constructs are mainly the tags used and could explain the reduced affinity observed for the N-terminally tagged TraI$_{pCU1}$ because adding additional residues at the N-terminus would likely disturb the active site. However, the differences are consistent also

with the C-terminally tagged TraI$_{pCU1}$, so we have no obvious explanation for the observed discrepancies between our experimental results and those of TraI$_{pCU1}$. We do note that our observations for TraI$_{pKM101}$ align well with published data for homologous proteins. We therefore conclude that TraI$_{pKM101}$ binds and processes DNA in a similar fashion as its homologs TrwC$_{R388}$ and TraI$_F$.

Although His-tags sometimes cause dimerization (Amor-Mahjoub et al, 2006; Singh et al, 2020), it is very uncommon that they promote any defined higher oligomerization. This led us to initially think that TraI formed tetramers in solution (Fig S2), but this turned out to be an artifact of the His-tag (Fig S1). However, an interesting feature in the EMSAs with full-length TraI, and to some extent its helicase domain, is the appearance of supershifts at higher protein concentrations (Figs 2 and S3). The exact same data were observed both with His-tagged TraI and TraI with the His-tag removed. This suggests that multiple copies of the protein can bind to the same DNA molecule. Support for this can be found in the literature as TraI$_{pKM101}$ was previously reported to self-interact in vivo (Li & Christie, 2020). In addition, TraI$_F$ was shown to bind *oriT* as a dimer where one subunit takes on a closed conformation, in which the helicase surrounds the DNA, and the other subunit takes on an open conformation, with the trans-esterase domain bound to *oriT*. These two conformations are proposed to be mutually exclusive as the directionality of the ssDNA was proposed to be inconsistent with the same protein binding with both the helicase and trans-esterase domain simultaneously (Ilangovan et al, 2017). Although there are significant differences between TraI$_F$ and TraI$_{pKM101}$ (Fig 1A), it is possible that TraI$_{pKM101}$ exhibits similar oligomerization when binding to DNA.

To understand the interaction between TraI and its substrate DNA on a mechanistic level, we solved the crystal structure of the trans-esterase domain in complex with its cognate *oriT* DNA. The DNA-bound structure of TraI-TE$_{pKM101}$ (Fig 4A) shows a protein fold

that can be described as a palm with the active site at the center and a thumb subdomain that is involved in holding the substrate DNA in place (Fig S4A). This structure is similar to the previously solved trans-esterase domains of its homologs (Guasch et al, 2003; Larkin et al, 2005; Boer et al, 2006) (Fig 4B). The highest structural homology is found with another member of the $MOB_{F11}$ subclade: $TrwC_{R388}$ (PDB: 2CDM, 50% identity, root mean square deviation (RMSD) = 1.7 Å over 276 residues). The structure is also similar to that of $TraI_F$, a $MOB_{F12}$ relaxase (PDB: 2A0I, 37% identity, RMSD = 2.3 Å over 269 residues) (Holm, 2020). The DNA-binding site is highly conserved between these proteins with the bound ssDNA making a characteristic U-turn (Carballeira et al, 2014). We modeled the divalent cation coordinated by the histidine triad as a $Mn^{2+}$ because it clearly gave the best result in the refinements (Fig S4B and C). However, we do not have any experimental data to conclude with certainty that it is indeed $Mn^{2+}$. The divalent cation found in the DNA-bound structure was likely retained during purification because no metals were added during purification or crystallization, whereas the apo structure was crystallized in the presence of both $Mg^{2+}$ and $Ca^{2+}$. Our activity assays were successfully performed in the presence of $Mg^{2+}$. Because TraI homologs, containing the same active sites, have been shown to be able to function with various divalent metals, we did not investigate the identity of the metal further (Boer et al, 2006; Nash et al, 2010).

Although the DNA-bound structures of the $MOB_F$ family relaxases are very similar, there is more variability between the structures in the absence of DNA. This probably reflects the greater flexibility of the structures in the absence of the ligand. The apo structure of $TraI-TE_{pKM101}$ presented here (Fig 5A) shows the thumb subdomain in an open conformation. Previously, this thumb subdomain could only be modeled in crystal structures with bound DNA (Guasch et al, 2003; Larkin et al, 2005; Boer et al, 2006). This thumb is also absent in the cryo-EM structure of $TraI_F$, which has DNA bound to the helicase domain (Ilangovan et al, 2017). The RMSD between our ligand-free and DNA-bound structures is 3.3 Å over 277 residues, which is indicative of the large molecular movement of up to 40 Å (Fig 5A and Video 1). It is likely that the open conformation is important for *oriT* DNA to access the active site. The exact degree of opening observed in our structure is dependent on crystal contacts that stabilize the thumb subdomain. In solution, the protein is very likely in an open–closed equilibrium, which is shifted strongly toward probably many different open states in the absence of cognate DNA.

We were interested whether the thumb movement would be possible in the context of the full-length protein. To gain insight, we compared our TraI-TE structure to $TraI_F$ (PDB: 5N8O) and the alphafold model of full-length TraI (Jumper et al, 2021; Terwilliger et al, 2022). Superimposing TraI-TE on to the full-length TraI alphafold model shows that the thumb subdomain is facing the helicase domain in the model, but no direct contacts are predicted between the thumb and the helicase domains (Fig S6A). Furthermore, the cryo-EM structure of $TraI_F$, which has DNA bound to the helicase domain, is missing the thumb subdomain (Fig S6B) (Ilangovan et al, 2017). This suggests that the thumb subdomain is flexible and exhibits an open-closed equilibrium also in $TraI_F$. Taken together, this makes it likely that the conformational change of the thumb observed for the trans-esterase domain in isolation is also fully possible in context of the full-length protein.

There are important differences between the structures of $TraI-TE_{pKM101}$ and the apo structure of $TraI-TE_{pCU1}$ (PDB: 3L6T) despite their 100% sequence identity (Fig 5B). The overall fold is similar, with a RMSD of 1.4 Å over 208 residues, but the entire C-terminal region of $TraI_{pCU1}$ is missing. This includes the thumb subdomain and the helix at the base of the thumb (α-helices 7–10 of the $TraI_{pKM101}$ DNA-bound structure). More importantly, α-helix 1, which contains the beginning of the conserved YY-X$_{(5-6)}$–YY motif that provides the catalytic tyrosine, mostly appears as a loop in the previously published $TraI_{pCU1}$ structure. As a consequence, it points far away from the active site, resulting in a >10 Å displacement compared with the positions in the homologs (Nash et al, 2010). Possibly, the displacement of α-helix 1 is related to the destabilization of the thumb subdomain of the protein. In contrast, both structures presented here, as well as other relaxase structures ($TrwC_{R388}$ PDB: 1S6M, $TraI_F$ PDB: 2Q7T), show interactions between the base of the thumb subdomain (α-helix 10) and the loop region following α-helix 1 (R274 to D25 in $TraI_{pKM101}$) (Fig S5B and C). In the DNA-bound structure of $TraI_{pKM101}$, an additional interaction was found between H278 (α-helix 10) and Y26 (the third Y of the YY-X$_{(5-6)}$–YY motif), which might further contribute to this stabilization.

To conclude, we have shown that $TraI_{pKM101}$ has a high affinity and specificity to its cognate *oriT* and can both nick and religate *oriT* in a similar manner as other $MOB_F$ family relaxases. The crystal structures of its trans-esterase domain ($TraI-TE_{pKM101}$) visualize the large conformational change the protein undergoes upon DNA binding. Both the biochemical function and structures of $TraI_{pKM101}$ reported here resemble those of the homologs $TrwC_{R388}$ and $TraI_F$ more closely than the previously published work on the identical $TraI-TE_{pCU1}$ (Nash et al, 2010). Our findings thus highlight the conserved mechanism of relaxases and the role they play in conjugation.

# Materials and Methods

### Plasmids

The full-length *traI* gene and shorter constructs encoding either the trans-esterase domain (TraI-TE) (residues 1–299) or the helicase domain (TraI-H) (residues 429–910) were PCR-amplified and cloned into p7XC3H and p7XNH3 via the FX cloning system (Geertsma & Dutzler, 2011) under the T7 promotor, with an N- or C-terminal 10-His-tag and a 3C protease sequence (see Table S1).

### Protein expression and purification

All TraI variants were produced in *E. coli* BL21 (DE3). The cells were grown in 1.5 liters TB medium at 37°C using a LEX (large-scale expression) bioreactor (Epiphyte3). Once the cultures reached an $OD_{600}$ between 1.0 and 1.5, the temperature was reduced to 18°C and expression was induced by adding 0.4 mM IPTG for ~16 h. Cells were centrifuged at 6,000*g*, and pellets were resuspended in resuspension buffer (50 mM Hepes-NaOH, pH 7.0, 300 mM NaCl, 15 mM Imidazole, 0.2 mM AEBSF, and ca 0.02 mg/ml DNase I). Resuspended cells were lysed in a Cell Disruptor (Constant

Systems) at 25 kPsi at 4°C, followed by centrifugation at 30,000$g$ for 30 min at 4°C. The cell lysate was then incubated for 1 h at 4°C with Ni-NTA resin (~2 ml/liters culture for TraI and TraI-H or ~4 ml/liters culture for TraI-TE) before transfer to a gravity flow column. The resin was subsequently washed with 10 column volumes (CV) of wash buffer (50 mM Hepes-NaOH, pH 7.0, 300 mM NaCl, 50 mM Imidazole, 0.2 mM AEBSF), followed by 10 CV wash buffer with 2 M LiCl and another 10 CV wash buffer. For experiments without tag-cleavage, TraI$_{His}$ was eluted with 5 CV elution buffer (50 mM Hepes-NaOH, pH 7.0, 300 mM NaCl, 500 mM Imidazole).

For experiments in which the tag was cleaved off, 3C PreScission Protease was added, while the protein was still bound to the IMAC resin, in an estimated ratio of 1:50–1:100 and incubated ~16 h at 4°C. The cleaved protein was then recovered by collecting the flow through.

All samples were diluted 3× with dilution buffer (25 mM Hepes-NaOH, pH 7.0, 50 mM NaCl) after Ni-NTA purification and loaded on a HiTraP Heparin HP (5 ml) column, equilibrated with Buffer A (25 mM Hepes-NaOH, pH 7.0, 150 mM NaCl) with a peristaltic pump. After loading the protein, the column was connected to an ÄKTA Pure (Cytiva) and washed with Buffer A for ~2 CV until absorbance at 280 nm was stable. All TraI variants were subsequently eluted via a salt gradient to 100% Buffer B (25 mM Hepes-NaOH, pH 7.0, 1.0 M NaCl) over 80 ml. Peak fractions with pure protein, as judged by a 260 nm/280 nm ratio of <0.8, were concentrated using Amicon Ultra Centrifugal filters with molecular weight cutoffs at 50 kD (TraI or TraI$_{His}$), 30 kD (TraI-H), or 10 kD (TraI-TE). Size-exclusion chromatography was performed in 25 mM Hepes-NaOH, pH 7.0, 300 mM NaCl on a Superdex 200 10/300 Gl Increase column, and fractions were analyzed on 12% SDS–PAGE stained with Coomassie blue.

## SEC-MALS

SEC-MALS was performed with 250 $\mu$l of protein sample at a minimum concentration of 1 mg/ml in 25 mM Hepes-NaOH, pH 7.0, 300 mM NaCl. Experiments were done on a Superdex 200 10/300 Gl Increase column or a Superose 6 10/300 Gl Increase column for TraI$_{His}$, on ÄKTA Pure (Cytiva) coupled to a light scattering (Wyatt Treas II) and refractive index (Wyatt Optilab T-Rex) detector. Data were collected and analyzed using Astra software (version 7.2.2; Wyatt Technology) as described by Some et al (2019). The molecular weight of the protein samples was calculated as an average from a minimum of 3 measurements and is reported with the standard deviation.

## DNA oligomers used for binding assays and crystallization

Single-stranded DNA oligomers were purchased from Eurofins Genomics, with and without the fluorescent label fluorescein isothiocyanate (FITC). The position of the label at the 5' or 3' end is indicated in Table S1, and a schematic of the different *oriT* DNA used in this study can be found in Fig 1B. Stock solutions of 100 mM were dissolved in MilliQ water, but all further dilutions were done in 25 mM Hepes-NaOH, pH 7.0, 300 mM NaCl. Oligomers were heated at 95°C for 5 min and cooled down to room temperature over at least 20 min before use.

## EMSA

Fluorescently labeled DNA (50 nM, Table S1) was incubated for 15 min with 0–1,600 nM of protein (see the Results section) in 25 mM Hepes-NaOH, pH 7.0, 300 mM NaCl at room temperature before adding 6× native loading dye (3× TBE, 30% glycerol, 0.125% bromophenol blue). The samples were resolved on a native gel (5% polyacrylamide, 0.75× TBE) in 0.75× TBE running buffer at 50 V for 90 min at 4°C. Gels were imaged on an Amersham Typhoon 5 scanner with excitation at 488 nm, using the Cy2 emission filter (515–535 nm).

## Nicking and religation assays

Nicking and religation assays were performed in 25 mM Hepes-NaOH, pH 7.0, 300 mM NaCl, 5 mM MgCl$_2$. Nicking assays contained 100 nM F-*oriT*57, religation assays (version 1) contained 100 nM F-*oriT*57 and 100 nM F-*oriT*20, and religation assays (version 2) contained 100 nM F-*oriT*35 and 100 nM non-fluorescent *oriT*57 (Fig 1B and Table S1). DNA oligomers were incubated with increasing concentrations of TraI in a shaking heatblock at 37°C and 300 rpm, for 1 h. The reactions were stopped by adding 2× stop solution (96% formamide, 20 mM EDTA, 0.1% bromophenol blue). The samples were boiled for 5 min at 95°C before loading on a denaturing gel (16% polyacrylamide, 7 M urea, 1× TBE). The gel was run at 50 V for 15 min followed by 100 V for 1 h at room temperature and then imaged on an Amersham Typhoon 5 scanner with excitation at 488 nm, using the Cy2 emission filter (515–535 nm).

## Crystallization and structure determination

Protein crystals were obtained with the sitting drop vapor diffusion method at 20°C. Substrate DNA, *oriT*11 (Fig 1B and Table S1), was added to TraI-TE, to yield a final concentration of 9 mg/ml TraI-TE and an equimolar amount of the DNA. This mixture was used for co-crystallization experiments, which yielded both the apo structure of TraI-TE and the DNA-bound structure. The apo structure crystallized in a 1:1 ratio with 10% wt/vol PEG 4000, 20% v/v glycerol, 0.03 M MgCl$_2$, 0.03 M CaCl$_2$, and 0.1 M MOPS/Hepes-Na, pH 7.5. The DNA-bound structure crystallized in a 1:1 ratio with 10% wt/vol PEG 20000, 20% vol/vol PEG MME 550, 0.03 M sodium nitrate, 0.03 M disodium hydrogen phosphate, 0.03 M ammonium sulfate, and 0.1 M MES/imidazole, pH 6.5. Crystals were flash-frozen in liquid nitrogen, without the addition of extra cryo-protectant. X-ray diffraction data were collected on beamline ID30B, at the ESRF, France. The data were processed using XDS (Kabsch, 2010; Monaco et al, 2013). Both crystals had the space group $P2_12_12_1$ and contained a single copy of the protein in the asymmetric unit. The phase problem was solved using molecular replacement with the trans-esterase domain of TraI$_{pCU1}$ (PDB: 3L6T) in PHENIX phaser (McCoy et al, 2007; Nash et al, 2010). The structures were built in Coot (Emsley & Cowtan, 2004) and refined at 2.1 Å (DNA-bound structure) and 1.7 Å (apo structure) in PHENIX refine (Afonine et al, 2012), to R$_{work}$/R$_{free}$ values of 22.5/26.9% and 17.1/19.1%, respectively. Further refinement statistics can be found in Table S2.

## Data Availability

Atomic coordinates and structure factors of both the apo and DNA-bound structures of TraI have been deposited with the Protein Data Bank (PDB: 8A1B and 8A1C).

## Supplementary Information

## Acknowledgements

A pET15b plasmid containing the pKM101 TraI gene that we used for further cloning in *E. coli* was generously donated by Prof. Peter J Christie, whom we also thank for fruitful discussions regarding the project. We acknowledge the MAX IV Laboratory for time on Beamline BioMax under Proposal 20180236. Research conducted at MAX IV, a Swedish national user facility, is supported by the Swedish Research council under contract 2018-07152, the Swedish Governmental Agency for Innovation Systems under contract 2018-04969, and Formas under contract 2019-02496. We also acknowledge the synchrotrons Swiss Light Source (Paul Scherrer Institute, Switzerland) for time at beamline PX1 and the ESRF (France) for time at beamlines ID23 and ID30. This work was supported by grants from the Swedish Research Council (2016-03599), Knut and Alice Wallenberg Foundation, and Kempestiftelserna (SMK-1762 and SMK-1869) to RP-A Berntsson.

### Author Contributions

A Breidenstein: conceptualization, formal analysis, validation, investigation, visualization, methodology, and writing—original draft, review, and editing.
J ter Beek: conceptualization, supervision, project administration, and writing—original draft, review, and editing.
RP-A Berntsson: conceptualization, data curation, formal analysis, supervision, funding acquisition, investigation, visualization, project administration, and writing—original draft, review, and editing.

### Conflict of Interest Statement

The authors declare that they have no conflict of interest.

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
