## [Reviewer comments · Life Science Alliance]

Life Science Alliance

Structural and functional characterization of Tral from pKM101 reveals basis for DNA processing

Annika Breidenstein, Josy ter Beek, and Ronnie Bertsson

DOI: <https://doi.org/10.26508/lsa.202201775>

Corresponding author(s): *Ronnie Bertsson, Umeå University*

Review Timeline:

Submission Date:	2022-10-19
Editorial Decision:	2022-12-13
Revision Received:	2022-12-21
Editorial Decision:	2023-01-09
Revision Received:	2023-01-10
Accepted:	2023-01-10

Scientific Editor: Novella Guidi

Transaction Report:

December 13, 2022

Re: Life Science Alliance manuscript #LSA-2022-01775

Dr. Ronnie P-A Berntsson
Umeå University
Department of Medical Biochemistry and Biophysics
Umeå 901 87
Sweden

Dear Dr. Berntsson,

Thank you for submitting your manuscript entitled "Structural and functional characterization of Tral from pKM101 reveals basis for DNA processing" to Life Science Alliance. The manuscript was assessed by expert reviewers, whose comments are appended to this letter. We invite you to submit a revised manuscript addressing the Reviewer comments.

Thank you for this interesting contribution to Life Science Alliance. We are looking forward to receiving your revised manuscript.

Sincerely,

B. MANUSCRIPT ORGANIZATION AND FORMATTING:

Reviewer #1 (Comments to the Authors (Required)):

This manuscript describes biochemical study of a relaxase Tral from pKM101 involved in T4SS and crystal structures of the trans-esterase domain (TE) of Tral in DNA-bound and DNA-free forms. The structure of TE from pKM101 is essentially the same as those of the homologous proteins. However, this study firstly visualizes the structure of the thumb sub-domain of the TE in the absence of DNA and the conformational change of the sub-domain upon DNA-binding. In addition, the authors characterize the DNA binding and re-ligation activities of TE, helicase domain, and full-length Tral using recombinant proteins without His-tag. These results provide valuable insight into mechanism underlying DNA binding and re-ligation of Tral. Experiments were nicely conducted and manuscript was well organized. However, the reviewer has several concerns shown below.

(1) The authors considered that Mn²⁺ was bound to the histidine triad of TE based on the electron density. The authors should show the electron density map and should explain in detail of the reason why the authors excluded possibility of other ions such as Zn²⁺. If the authors observed anomalous data, it would be useful to identify the metal ion. Alternatively, difference map or B-factor would be useful.

Considering the coordination number (3 or 4?) and the ligand residues, the metal ion may be Zn²⁺. Are there water molecules coordinating with the metal ion?

(2) The author estimated an apparent K_d as 200 nM. The authors should describe in detail how they estimated the value.

(3) Line 246-255

Some descriptions of structural determination are duplicated in the materials and method section and so should be shortened here.

(4) It may be better to use "subdomain" for thumb or palm rather than "domain", because these are located in the trans-esterase domain.

(5) Figure 4

The authors should indicate thumb and palm in a structural presentation.

Minor points:

Line 63

The review paper may not include structural information of relaxases. The authors should confirm if the ref.5 is appropriate.

Line 98

The authors should provide a base compound (NaOH or KOH?) used to adjust pH.

Line 173, P212121

P should be in italics. 1 should be subscripted.

Line 225

Is oriTDNA about oriT35?

Line 232

Is 4A a mistake for 3A?

Line 249, P212121

P should be in italics. 1 should be subscripted.

Line 350

pCU1 in TralpCU1 should be subscripted.

Line 405

"motive" should be "motif".

Line 554

Mg²⁺ should be Mn²⁺, if the authors consider that Mn²⁺ is bound to the TE.

Reviewer #2 (Comments to the Authors (Required)):

Conformational change in relaxase proteins during the initiation phase of conjugative DNA transfer is key to controlling conjugation. Recent efforts in structural biology have sought to capture the open and closed states of relaxase proteins on single-stranded plasmid DNA and extract mechanistic insights regarding coordination of their dual trans-esterase / helicase activities and control of the overall DNA transfer process.

In this report, Breidenstein et al isolate full length Tral protein of the pKM101 plasmid as well as two truncated variants: the N-terminal transesterase domain, Tral-TE and the helicase domain, Tral-H. The authors present both the DNA-bound structure of Tral-TEpKM101 and the apo structure of this variant. Protein purification and approaches taken to confirm biochemical activities ascribed to relaxases appear sound.

The authors' finding that a large conformational movement concomitant to folding the C-terminal thumb region of the truncated Tral-TE protein occurs upon ssDNA binding is important. How do the authors envisage that transition occurring in the full length Tral protein where the alpha helices involved are tethered to the larger helicase domain? In the context of the full length Tral of F-family plasmids that domain is firmly anchored 3' to nic in its closed helicase confirmation (Ilangovan, A., 2017). Breidenstein et al also show that TralpKM101 formed monomers in solution and may form dimers when bound to oriT. Do the authors hypothesize that both helicase and TE closed states can be present in an active monomer? Structural clashes surrounding the nic site may impede that possibility.

It is also conceivable that two monomers bind at oriT to enable both the transesterase and the helicase conformers to form and carry out their respective DNA processing reactions.

The authors should comment on this range of possibilities. It would be helpful if they would also discuss their observations in a larger context considering both the trans-esterase and DNA helicase activities of the Tral protein.

Minor points:

The authors do a good job describing the complex process of T4 secretion in generally understandable terms. Nonetheless, I think readers new to the field would benefit if basic concepts would be presented more fully, such as the various essential components that comprise conjugative T4SS (extracellular pilus, multicomponent membrane spanning transporter, coupling protein and substrate: ssDNA). More comprehensive citations throughout the manuscript would be a help. As it now stands, readers are left on their own to search for relevant literature. This is the case if the reader seeks information regarding T4S generally, and specifically to gain information concerning pKM101 and pCU1 plasmids. This could be improved.

line 232, 240 and 244: these data are shown in Figure 3 not 4.

line 106: please clarify what 5ml/CV means for this elution. It would seem to be a dilution of the high imidazole containing buffer???

line 184. were the "functional domains" predicted or defined biochemically? if the latter please add citation.

line 251. Please cite publication

First of all, we would like to thank the reviewers for taking their time to critically read and comment on our manuscript. Please find our point-by-point response to all reviewer comments below (reviewer comments in black, our responses in red text).

Reviewer #1 (Comments to the Authors (Required)):

This manuscript describes biochemical study of a relaxase TraI from pKM101 involved in T4SS and crystal structures of the trans-esterase domain (TE) of TraI in DNA-bound and DNA-free forms. The structure of TE from pKM101 is essentially the same as those of the homologous proteins. However, this study firstly visualizes the structure of the thumb sub-domain of the TE in the absence of DNA and the conformational change of the sub-domain upon DNA-binding. In addition, the authors characterize the DNA binding and re-ligation activities of TE, helicase domain, and full-length TraI using recombinant proteins without His-tag. These results provide valuable insight into mechanism underlying DNA binding and re-ligation of TraI. Experiments were nicely conducted and manuscript was well organized. However, the reviewer has several concerns shown below.

(1) The authors considered that Mn²⁺ was bound to the histidine triad of TE based on the electron density. The authors should show the electron density map and should explain in detail of the reason why the authors excluded possibility of other ions such as Zn²⁺. If the authors observed anomalous data, it would be useful to identify the metal ion. Alternatively, difference map or B-factor would be useful.

Considering the coordination number (3 or 4?) and the ligand residues, the metal ion may be Zn²⁺. Are there water molecules coordinating with the metal ion?

What we do observe is electron density at the active site which is much larger than a water. We tried refinement with the common divalent cations (Zn²⁺, Mn²⁺, Mg²⁺, Ca²⁺), and since Mn²⁺ gave the best result we modelled this in the structure. However, it could very well be that there is a mixture of metals bound in the crystal structure. Especially in the apo structure, there is additional density for two water molecules that are coordinated by the metal ion. The density for these waters are less clear in the DNA bound structure.

We did not mean to argue that Mn²⁺ *per se* is the metal used by TraI *in vivo*, and that no other ions could be used and apologize if that was how it came across. Homologous proteins have indeed been shown to be able to use most divalent cations to catalyze the nicking reaction in a similar manner, and everything points to this being true for TraI as well, since our activity assays were performed in the presence of Mg²⁺ (and gave similar results also when we first stripped the protein using EDTA). Since it has been clearly shown that TraI homologs can utilize different metal ions and still be perfectly functional (and the active site is highly conserved), we did not see the need to further characterize the metal preference of TraI.

We have now modified the manuscript to make this clearer; *I.e.* that we have modelled it as a Mn²⁺, but that it could be another metal and highlighting that close homologs have been shown to be functional with several different divalent cations. (lines 189-190, 317-325) We have also added figures of the electron density at the active site, including the metal ion, as new Fig. S4 B & C.

(2) The author estimated an apparent K_d as 200 nM. The authors should describe in detail how they estimated the value.

The bands in our EMSAs are difficult to quantify because of smearing. Therefore, we have not tried to give an exact value for the affinity. The apparent K_d that we describe is the concentration where half of the DNA is bound by TraI. In Fig. 2A, somewhat more than half of the DNA (ca 55-60%, depending on the quantification) is upshifted in the lane with 200 nM protein and we therefore concluded that the K_d is below 200 nM. We have changed this in the results (line 133) and clarified this in the discussion (line 260-262).

(3) Line 246-255

Some descriptions of structural determination are duplicated in the materials and method section and so should be shortened here.

We have modified the text accordingly.

(4) It may be better to use "subdomain" for thumb or palm rather than "domain", because these are located in the trans-esterase domain.

We completely agree with the reviewer and have now modified this accordingly throughout the manuscript.

(5) Figure 4

The authors should indicate thumb and palm in a structural presentation.

Thank you for the good suggestion. We have now added a new supplementary figure (Fig. S4A) that indicates the palm and thumb subdomains.

Minor points:

Line 63

The review paper may not include structural information of relaxases. The authors should confirm if the ref.5 is appropriate.

Zechner et al (2017) contains structural information of relaxases, and is appropriate to have along as a reference here.

Line 98

The authors should provide a base compound (NaOH or KOH?) used to adjust pH.

We apologize that this was missing. NaOH was used to adjust the pH. This information has been added to the manuscript.

Line 173, P212121

P should be in italics. 1 should be subscripted.

Thank you. This has been fixed.

Line 225

Is oriTDNA about oriT35?

This observation is valid for both oriTs tested (oriT35 and oriT57). We have now clarified this in the text.

Line 232

Is 4A a mistake for 3A?

Thank you for catching this mistake. We have now adjusted the figure numbers accordingly.

Line 249, P212121

P should be in italics. 1 should be subscripted.

Done.

Line 350

pCU1 in TraIpCU1 should be subscripted.

Done

Line 405

"motive" should be "motif".

Done

Line 554

Mg²⁺ should be Mn²⁺, if the authors consider that Mn²⁺ is bound to the TE.

Thank you for catching this mistake. Has now been changed.

Reviewer #2 (Comments to the Authors (Required)):

Conformational change in relaxase proteins during the initiation phase of conjugative DNA transfer is key to controlling conjugation. Recent efforts in structural biology have sought to capture the open and closed states of relaxase proteins on single-stranded plasmid DNA and extract mechanistic insights regarding coordination of their dual trans-esterase / helicase activities and control of the overall DNA transfer process.

In this report, Breidenstein et al isolate full length TraI protein of the pKM101 plasmid as well as two truncated variants: the N-terminal transesterase domain, TraI-TE and the helicase domain, TraI-H. The authors present both the DNA-bound structure of TraI-TEpKM101 and the apo structure of this variant. Protein purification and approaches taken to confirm biochemical activities ascribed to relaxases appear sound.

The authors' finding that a large conformational movement concomitant to folding the C-terminal thumb region of the truncated TraI-TE protein occurs upon ssDNA binding is important. How do the authors envisage that transition occurring in the full length TraI protein where the alpha helices involved are tethered to the larger helicase domain?

Also in the context of the full-length protein the thumb subdomain will very likely have the possibility to undergo this large conformational change. This hypothesis is supported by the AlphaFold model of the full-length TraI, which indicates that there are no direct contacts between the thumb subdomain and the helicase domain. Secondly, in the EM structure of the F-plasmid TraI, the thumb subdomain is not present in the structure, likely due to it being highly flexible. Thus, again indicating its flexibility. Superimposing our DNA-bound TraI-TE structure to the F-plasmid TraI shows that there are no steric clashes (except for one side-chain in a flexible loop region) between the thumb subdomain and

the active helicase domain. Taken together, this strongly indicates that the thumb subdomain has the possibility to change its conformation as observed in our structures of the TE-domain only.

We have added a section about this in the manuscript (lines 341-351) and new Fig. S6 (showing our TraI structure superimposed on either the AlphaFold-model or the F-plasmid TraI).

In the context of the full length TraI of F-family plasmids that domain is firmly anchored 3' to nic in its closed helicase confirmation (Ilangovan, A., 2017).

Breidenstein et al also show that TraIpKM101 formed monomers in solution and may form dimers when bound to oriT. Do the authors hypothesize that both helicase and TE closed states can be present in an active monomer? Structural clashes surrounding the nic site may impede that possibility. It is also conceivable that two monomers bind at oriT to enable both the transesterase and the helicase conformers to form and carry out their respective DNA processing reactions.

Similar to the situation in TraI from the F-plasmid, it is likely that the DNA bound states of the TE domain and the helicase domain are mutually exclusive, which is to some extent supported by comparing the AlphaFold model of TraI with the cryo-EM structure of TraI from the F-plasmid. However, we cannot say this for certain based on the available data and have therefore preferred to keep this as speculation.

We have modified the discussion at lines 296-304 to make this point.

The authors should comment on this range of possibilities. It would be helpful if they would also discuss their observations in a larger context considering both the trans-esterase and DNA helicase activities of the TraI protein.

As mentioned above, we have now added parts to the discussion where we discuss these different aspects of the protein function.

Minor points:

The authors do a good job describing the complex process of T4 secretion in generally understandable terms. Nonetheless, I think readers new to the field would benefit if basic concepts would be presented more fully, such as the various essential components that comprise conjugative T4SS (extracellular pilus, multicomponent membrane spanning transporter, coupling protein and substrate: ssDNA). More comprehensive citations throughout the manuscript would be a help. As it now stands, readers are left on their own to search for relevant literature. This is the case if the reader seeks information regarding T4S generally, and specifically to gain information concerning pKM101 and pCU1 plasmids. This could be improved.

We have now added more information about T4SSs in general to the introduction (lines 47-59), and have also added more citations.

line 232, 240 and 244: these data are shown in Figure 3 not 4.

Thank you for catching this mistake, this has now been addressed.

line 106: please clarify what 5ml/CV means for this elution. It would seem to be a dilution of the high imidazole containing buffer???

We apologize for the typo, it was meant to say 5 CV. This has now been fixed.

line 184. were the "functional domains" predicted or defined biochemically? if the latter please add citation.

The domain boundaries were predicted using PsiPred (this was done before the arrival of AlphaFold). This has now been added to the text (including the citation to PsiPred).

line 251. Please cite publication

This line is now removed, as requested by reviewer #1. We have however cited the publication of this structure in the materials and methods section.

January 9, 2023

RE: Life Science Alliance Manuscript #LSA-2022-01775R

Dr. Ronnie P-A Berntsson
Umeå University
Department of Medical Biochemistry and Biophysics
Umeå 901 87
Sweden

Dear Dr. Berntsson,

Thank you for submitting your revised manuscript entitled "Structural and functional characterization of Tral from pKM101 reveals basis for DNA processing". We would be happy to publish your paper in Life Science Alliance pending final revisions necessary to meet our formatting guidelines.

- please add a conflict of interest statement to the main manuscript text
- please add a figure callout for Figure S5 B,C to the main manuscript text

Figure Check:

- Figure S1 has panels A-C, but panel C is not mentioned in the figure legend. Please add it.

A. FINAL FILES:

B. MANUSCRIPT ORGANIZATION AND FORMATTING:

Sincerely,

Reviewer #1 (Comments to the Authors (Required)):

The authors have carefully revised the original manuscript in accordance with the reviewer's comments or provided appropriate explanations for concerns by the reviewer. Therefore, the reviewer has no additional comments or concerns on the revised manuscript, and considers this manuscript is now ready for publication.

Reviewer #2 (Comments to the Authors (Required)):

The revision submitted by Breidenstein, et al. improved the text, made the topic of bacterial type IV secretion more generally accessible and improved the data presentation. I also appreciate the extended discussion. The authors have placed their findings with the TE domain in the context of the larger protein and include the wider context of plasmid DNA processing during conjugation. The paper's message was strengthened as a result. In my view, this study is ready for publication.

January 10, 2023

RE: Life Science Alliance Manuscript #LSA-2022-01775RR

Dr. Ronnie P-A Berntsson
Umeå University
Department of Medical Biochemistry and Biophysics
Umeå 901 87
Sweden

Dear Dr. Berntsson,

Thank you for submitting your Research Article entitled "Structural and functional characterization of Tral from pKM101 reveals basis for DNA processing". It is a pleasure to let you know that your manuscript is now accepted for publication in Life Science Alliance. Congratulations on this interesting work.

DISTRIBUTION OF MATERIALS:

Again, congratulations on a very nice paper. I hope you found the review process to be constructive and are pleased with how the manuscript was handled editorially. We look forward to future exciting submissions from your lab.

Sincerely,
